# Enhancement of SARS-CoV-2 mRNA Vaccine Efficacy through the Application of TMSB10 UTR for Superior Antigen Presentation and Immune Activation

**DOI:** 10.3390/vaccines12040432

**Published:** 2024-04-17

**Authors:** Xiaoyan Ding, Yuxin Zhou, Jiuxiang He, Jing Zhao, Jintao Li

**Affiliations:** 1College of Basic Medicine, Third Military Medical University, Chongqing 400038, China; xiaoyan.ding@med.uni-muenchen.de (X.D.); 13795815231@163.com (Y.Z.); chqhjx@126.com (J.H.); jingzhao0815@163.com (J.Z.); 2Department of Pediatrics, Ludwig-Maximilians University of Munich, 80337 Munich, Germany

**Keywords:** UTR, SARS-CoV-2, mRNA vaccine, antigen-presenting cells

## Abstract

The development of effective vaccines against SARS-CoV-2 remains a critical challenge amidst the ongoing global pandemic. This study introduces a novel approach to enhancing mRNA vaccine efficacy by leveraging the untranslated region (UTR) of TMSB10, a gene identified for its significant mRNA abundance in antigen-presenting cells. Utilizing the GEO database, we identified TMSB10 among nine genes, with the highest mRNA abundance in dendritic cell subtypes. Subsequent experiments revealed that TMSB10’s UTR significantly enhances the expression of a reporter gene in both antigen-presenting and 293T cells, surpassing other candidates and a previously optimized natural UTR. A comparative analysis demonstrated that TMSB10 UTR not only facilitated a higher reporter gene expression in vitro but also showed marked superiority in vivo, leading to enhanced specific humoral and cellular immune responses against the SARS-CoV-2 Delta variant RBD antigen. Specifically, vaccines incorporating TMSB10 UTR induced significantly higher levels of specific IgG antibodies and promoted a robust T-cell immune response, characterized by the increased secretion of IFN-γ and IL-4 and the proliferation of CD4^+^ and CD8^+^ T cells. These findings underscore the potential of TMSB10 UTR as a strategic component in mRNA vaccine design, offering a promising avenue to bolster vaccine-induced immunity against SARS-CoV-2 and, potentially, other pathogens.

## 1. Introduction

The ongoing evolution of global infectious diseases underscores the increasing necessity and criticality of vaccine development. As an emerging vaccine technology, mRNA vaccines, which utilize the intrinsic mechanisms of human cells to produce specific pathogen antigens, provide a new direction for vaccine development. Properly designed mRNA sequences can enhance the targeting and stability of mRNA vaccines, thereby improving their efficacy and durability [1]. Traditional non-replicating mRNA vaccines typically comprise a 5′ cap structure, a 5′-untranslated region (UTR), an open reading frame (ORF) encoding the target antigen, a 3′-UTR, and a poly(A) tail structure [2]. UTRs are critical factors that regulate mRNA stability and translation efficiency, playing significant roles in cellular transcription and translation processes [3]. In mRNA vaccine design, the selection of UTRs directly impacts the expression levels of the vaccine and may also modulate its immunogenicity [4,5,6]. UTRs may be derived from naturally occurring UTRs of highly expressed genes, such as the UTRs from α- and β-globin genes, as demonstrated in the selection made for the SARS-CoV-2 vaccine BNT162b2 [7]. Alternatively, UTRs from the pathogen itself can be employed, or advantageous UTRs can be identified through systematic evolution and high-throughput screening, facilitated by artificial intelligence (AI) tools [8,9]. However, considering the variability in UTR efficacy across different cell types, there arises a critical need for the development of alternative UTR sequences optimized for specific application needs and intended cellular targets [10,11].

The efficacy of the adaptive immune response to pathogens and vaccines is intricately linked to the activation and functionality of dendritic cells (DCs). Serving as sentinel cells, DCs undertake the critical task of surveilling the body for pathogens and vaccine components. Upon encountering these entities, DCs efficiently process antigens and, subsequently, migrate to adjacent lymph nodes via the lymphatic system. Within the lymph nodes, DCs orchestrate the presentation of processed antigens to other immune cells, thereby orchestrating the initiation of a targeted and adaptive immune response tailored to the encountered threat [12]. DCs are a population of specialized antigen-presenting cells (APCs) consisting of different subtypes, among which immature DCs exhibit a high migratory capacity, while mature DCs can effectively activate naïve T cells, playing a crucial role in initiating, modulating, and maintaining immune responses [13]. Following the intramuscular administration of lipid nanoparticle (LNP)-mRNA vaccines, neutrophils, monocytes, and dendritic cells (DCs) are recruited to the injection site, where they produce inflammatory secretory factors, including chemokines and other inflammatory mediators. These factors facilitate the extravasation of immune cells. Subsequently, antigen-presenting cells (APCs), especially DCs, effectively uptake and translate the LNP-mRNA, migrate to the nearby lymph node, and enhance antigen presentation, thereby promoting an adaptive immune response [14,15]. Targeting the antigen to DCs increases antibody levels [16]. Dendritic cells in the peripheral blood such as plasmacytoid dendritic cells produce large amounts of type 1 interferon in response to microbial, especially viral, infections and stimulate the corresponding T-cell response [17]. These dendritic cells can be transported to various tissues and organs in the human body through blood circulation [18]. Therefore, we hypothesized that optimizing the mRNA sequence to enhance its stability and translation efficiency in peripheral blood dendritic cells could improve the efficacy and durability of mRNA vaccines. In this study, we conducted a screening of untranslated regions (UTRs) derived from highly mRNA-abundant genes in peripheral blood dendritic cells using a bioinformatics analysis. We discovered that the expression of reporter genes in peripheral blood dendritic cells was amplified when utilizing both the 5′UTR and 3′UTR of TMSB10. Subsequently, we employed LNP encapsulation to deliver the SARS-CoV-2 RBD antigen modified with the UTR of TMSB10 and immunized mice via the intramuscular route. Our findings demonstrated that the modified RBD antigen enhanced antigen-specific humoral and T-cell immunity. This investigation highlights the potential of the original UTR of TMSB10 to enhance the immunogenicity of mRNA vaccines, offering promise for diverse applications in the development of viral mRNA vaccines.

## 2. Materials and Methods

### 2.1. Molecular Cloning and mRNA Synthesis

The plasmid sequence was designed using Gaussia luciferase (GLuc) reporter gene as the open reading frame (ORF) whose signal peptide was replaced by tissue plasminogen activator (tPA) signal peptide, with the 5′UTR and 3′UTR derived from different genes, followed by the addition of the T7 promoter sequence before the 5′UTR (sequences showed in Appendix A). The plasmids were synthesized by GenScript (Nanjing, China) and cloned into pUC57 vector. mRNA was produced using T7 High-Yield RNA Transcription kit (Novoprotein, Shanghai, China) on linearized plasmids. Then, Cap 1 was added to the synthesized RNA by using the Cap 1 Capping System (Novoprotein, Shanghai, China) and Pseudo-UTP (Ψ-UTP, APExBIO, Houston, TX, USA) was fully substituted for UTP. The Poly(A) tails were added using *E. coli* Poly(A) Polymerase (Novoprotein, Shanghai, China).

### 2.2. Cell Culture and Transfection

The HEK 293T cell, DC2.4, or RAW264.7 cell lines were purchased from Biospes Company (Chongqing Biospes Co., Ltd., Chongqing, China). Human bronchial epithelial cell line, 16HBE14o- Human Bronchial Epithelial Cell Line (16HBE), was purchased from Sigma (Sigma-Aldrich, St. Louis, MA, USA). In vitro transient transfection of mRNA was conducted in HEK 293T cells, DC2.4, 16HBE, or RAW264.7 using Lipofectamine^®^ 3000 reagent (Invitrogen, Carlsbad, CA, USA) following the manufacturer’s instructions. Briefly, HEK 293T cells and RAW264.7 were cultured in DMEM (Gibco, Billings, MT, USA), while DC2.4 and 16HBE were cultured in RPMI 1640, supplemented with 10% FBS (Gibco, Billings, MT, USA) and 1% penicillin-streptomycin (Gibco, Billings, MT, USA). Cell dissociation was achieved using 0.25% TrypLE (Thermo Fisher Scientific, Waltham, MA, USA), and cells were seeded in 48-well plates at a density of 50,000 cells per well. After 18 h, the medium was replaced with 2% FBS-medium, and cells were transfected with mRNA (0.25 μg per well) using Lipofectamine 3000 Transfection Reagent. Supernatants were collected 24 h post-transfection and stored at −80 °C.

### 2.3. Gaussia Luciferase (GLuc) Assay In Vitro

In vitro evaluation of GLuc expression was conducted using the Secrete-Pair™ Gaussia Luciferase Assay Kit (GeneCopoeia, LF062, Rockville, MD, USA). After thawing the collected supernatant on ice, 10 μL of the supernatant was added to a black 96-well plate for measurement. A working solution was prepared according to the kit instructions (100 μL per well), and the chemiluminescence value was measured at 500 ms using a Varioskan Lux (Thermo Scientific, Waltham, MA, USA).

### 2.4. Preparation of Lipid-GLuc mRNA Nanoparticles

In brief, SM-102 (AVT, Shanghai, China), DSPC (AVT, Shanghai, China), cholesterol (AVT, Shanghai, China), and DMG-PEG2000 (AVT, Shanghai, China) were dissolved in ethanol at a molar ratio of 50:10:38.5:1.5. And the final concentrations were 26.33 mg/mL for SM-102, 5.86 mg/mL for DSPC, 2.79 mg/mL for DMG-PEG2000, and 11.04 mg/mL for cholesterol. The lipid solution was, then, prepared by mixing SM-102, DSPC, DMG-PEG2000, and cholesterol solution in a 1:1:1:1 volume ratio. The mRNA was solubilized in 50 mM citrate buffer (pH 4.0) with concentration of 0.17 mg/mL. Then, 1 mL of lipid and 3 mL of mRNA solution (N:P ratio of 6:1) are mixed at room temperature through a dual-channel syringe pump at a flow rate of 1:3 (2 mL/min:6 mL/min) through the SCARM Mixer (Shuogen Technology Co., Ltd., Dongguan, China) to form nanoparticles of approximately 100 nm in diameter. The calculation formula of the N:P ratio is as follows:

N provided by SM-102. The molar concentration of SM102 in lipid solution is 9.267 mmol/mL. And the P is from the phosphate group of the RNA bases. The average molecular weight of the RNA bases (324.5 g/mol) was used to calculate the P.

The amount of substance (*n*) N:*n*_(*N*)_ = C_*N*_ × V = 9.267 mmol/L × 1 mL = 9.267 × 10^−6^ mol

C_*N*_: the concentration of the SM102

The amount of substance P:*n*_(*P*)_ = (C_*P*_ × V) ÷ M = (0.17 mg/mL × 3 mL) ÷ 324.5 g/mol ÷ 1000 = 1.545 × 10^−6^ mol

C_*P*_: the concentration of the mRNA
N:P ratio = *n*_(*N*)_ ÷ *n*_(*P*)_ = 9.267 × 10^−6^ mol ÷ 1.545 × 10^−6^ mol ≈ 6:1.

LNP-encapsulated mRNA samples were dialyzed against PBS (pH 7.4) in dialysis bags (Viskase, Lombard, IL, USA) for 24 h and stored at 4 °C until use. Encapsulation efficiency was measured using the Quant-iT RiboGreen RNA Assay Kit (Invitrogen, Carlsbad, CA, USA) with a Varioskan Lux (Thermo Scientific, Waltham, MA, USA).

### 2.5. Gaussia Luciferase (GLuc) Assay In Vivo

LNPs for in vivo imaging were formulated with mRNA encoding Gaussia luciferase (GLuc). The formulated LNPs were administered intramuscularly to mice at a dose of 5 μg of GLuc. Six hours later, tail blood samples were collected from the mice, and the fluorescence value of whole blood was measured using the Secrete-Pair™ Gaussia Luciferase Assay Kit (GeneCopoeia, LF062, Rockville, MD, USA), following the instructions provided. Following blood collection, coelenterazine was injected intraperitoneally immediately and incubated for 5 min. Luciferase expression in different organs was confirmed using an IVIS (PerkinElmer, Waltham, MA, USA).

### 2.6. Immunization and Detection of Antigen-Specific Antibodies in Mice

Female BALB/c mice were obtained from the Animal Center of the Third Military Medical University, and the animal experiments were ethically approved by the Laboratory Animal Welfare and Ethics Committee of the Third Military Medical University (Approval No. AMUWE20201373). For vaccinations, groups of 6- to 8-week-old BALB/c mice were immunized on days 0 and 7. The mRNA vaccine and the empty carrier control group were administered via intramuscular injection. Each dose of the mRNA vaccine contained 15 μg of mRNA, approximately 100 μL for two thighs. The empty LNP (PBS-LNP) was used as the control. After 14 days post-immunization, 100 μL of blood was collected from the mouse tail vein, followed by centrifugation at 4 °C for 10 min at 3000 rpm to isolate the serum. The serum was subsequently stored at −80 °C until further analysis. The Mice anti-SARS-CoV-2 (S-RBD) IgG ELISA Kit (FineTesT, Wuhan, China) was employed to measure antibody concentrations. Absorbance at 450 nm was measured using Varioskan Lux (Thermo Scientific, Waltham, MA, USA). 

### 2.7. Enzyme-Linked Immunospot (ELISPOT) Assays

Cellular immune responses in vaccinated mice were evaluated using IFN-γ and IL-4 pre-coated ELISPOT kits (MabTech, Hamburg, Germany). According to the instructions, plates were first blocked with 10% FBS in RPMI 1640 (Thermo Fisher Scientific, Waltham, MA, USA) and incubated for 30 min. Then, 1,000,000 splenocytes cells of the immunized mice were seeded in each well and inoculated along with a pool of SARS-CoV-2 RBD peptides (2 mg/mL of each peptide) [19]. After incubation at 37 °C with 5% CO_2_ for 36 h, the plates were washed with wash buffer, and biotinylated anti-mouse IFN-γ and IL-4 antibodies were added to each well, followed by a 2 h incubation at room temperature. Subsequently, AEC substrate solution was added, and, after air-drying, the plates were read using an automated ELISPOT reader (AID Classic EliSpot Reader, Strassberg, Germany). The numbers of spot-forming cells (SFC) per 1,000,000 cells were calculated.

### 2.8. Flow Cytometry Analyses for Mouse Splenocytes

Evaluation of T cell proliferation in immunized mice was conducted using a FACSCalibur flow cytometer (BD Biosciences, Milpitas, CA, USA). Briefly, a total of 1,000,000 mouse splenocytes were stimulated with the SARS-CoV-2 RBD peptides pool (4 μg/mL of each peptide) for 4 h at 37 °C with 5% CO_2_. Brefeldin A (1 mg/mL, BD Sciences, Milpitas, CA, USA) was, then, added to the splenocytes and incubated for an additional 4 h. The splenocytes were washed twice with the PBS, and then stained with fluorescently conjugated antibodies to CD3-FITC (BD Pharmingen, San Diego, CA, USA), CD4 (PerCP-Cyanine5.5, BD Pharmingen, San Diego, CA, USA), CD8-PE-Cyanine7 (BD Pharmingen, San Diego, CA, USA), CD44-APC (BD Pharmingen, San Diego, CA, USA), and CD62L-PE (BD Pharmingen, San Diego, CA, USA). Zombie NIR™ Fixable Viability Kit (BioLegend, San Diego, CA, USA), whose stain has similar emission to APC/Cy7, was used to evaluate live or dead status of mammalian cells. Data were analyzed using FlowJo software (Version 10.8.1).

### 2.9. Data Analysis

All statistical analyses were conducted using GraphPad Prism V8.0.2 software. The Student’s *t*-test and one-way ANOVA with multiple comparisons tests were employed for statistical comparisons between groups. A *p*-value ≤ 0.05 was considered indicative of a significant difference between groups.

## 3. Results

### 3.1. Screening and Preliminary Application of the TMSB10-UTR

In this study, we conducted an analysis to rank gene mRNA abundance in dendritic cells, encompassing both plasmacytoid dendritic cells (pDCs) and myeloid dendritic cells (mDCs) through utilizing the human peripheral blood mononuclear cell sequencing dataset GSE94820 from the Gene Expression Omnibus (GEO) database. The top 20 genes with the highest abundance for expression in these cell types were selected (Figure 1a–d), and, among them, nine genes—MTRNR2L2, HLA-DRA, HLA-DRB1, TMSB4X, ACTB, TMSB10, B2M, FTL, and FTH1—were consistently expressed in all dendritic cells’ subtypes (Figure 1e). The high mRNA abundance of these genes implies their significance in antigen-presenting cells, suggesting that their mRNAs have a high stability in antigen-presenting cells. 

To examine the regulation of the target gene expression by the UTRs of these nine genes, we inserted the UTRs of these genes into a modified reporter gene Gaussian luciferase (TPA-GLuc) and made nine constructs. Then, we produced nine mRNAs with the Cap 1 and Poly(A) tail based on these constructs (Figure 2a). Then, the nine mRNAs were transfected into 293T and DC2.4 cells (a model for antigen-presenting cells). After 24 h, we found that the UTRs of TMSB4X, B2M, and TMSB10 exhibited the significant ability to enhance TPA-GLuc expression in antigen-presenting cell types (namely, the constructs TMSB4X-UTR-Gluc, B2M-UTR-GLuc, and TMSB10-UTR-GLuc). Notably, the UTR of TMSB10 demonstrated the highest potency in regulating the reporter gene expression within DC2.4 cells (Figure 2b).

### 3.2. TMSB10-UTR Enhances Target Gene Expression in Antigen-Presenting Cells and In Vivo

To evaluate the effectiveness of these highly expressed UTRs in comparison to previously reported optimized natural UTR, we utilized the R27-UTR as described by Zeng et al. (2020), which has undergone optimization and demonstrates an expression efficiency comparable to other optimal constructs, including CYBA and AG + G (modified alpha globin UTRs).

Comparative analyses were conducted between the UTR of R27 and the UTRs of TMSB4X, B2M, and TMSB10 in regulating the GLuc gene expression in both antigen-presenting cells. Our results demonstrated that TMSB10-UTR-GLuc exhibited higher expression levels in DC2.4 cells compared with R27-UTR-Gluc. However, the expression with TMSB10-UTR-GLuc in mouse macrophage line RAW264.7 cells showed the same level as R27-UTR-GLuc (Figure 3b). This finding suggests that TMSB10-UTR-GLuc exhibits a superior expression in DC cells compared to R27-UTR-GLuc. Following this, GLuc-LNP with different UTRs (namely, TMSB10-UTR-Gluc and R27-UTR-GLuc) were intramuscularly administered to mice, and the bioluminescence values in whole blood were evaluated after 6 h. Remarkably, the expression of TMSB10-UTR-GLuc exhibited a notable increase compared to R27-UTR-GLuc in the blood (Figure 3c). This demonstrates that the UTRs screened from antigen-presenting cells from peripheral blood indeed enable a high expression of genes in peripheral blood. Upon dissecting the mice, we observed that TMSB10-UTR exhibited higher expression levels in the spleen and liver compared to R27-UTR-GLuc (Figure 3d). These findings underscore the substantial promise of TMSB10-UTR in governing the predominant target gene expression in animal antigen-presenting cells, offering an effective UTR selection strategy for mRNA vaccine design.

### 3.3. TMSB10-UTR Enhances SARS-CoV-2 mRNA Vaccine Efficacy

To validate the efficacy of TMSB10-UTR for mRNA vaccine applications, we proceeded to design mRNA sequences incorporating both R27-UTR and TMSB10-UTR. These sequences were developed using the SARS-CoV-2 Delta variant strain RBD (B.1.617.2, GenBank: OK091006.1) as the antigen, along with TPA as the signal peptide. Subsequently, we synthesized the mRNA in vitro, followed by the formulation of mRNA-LNP vaccines. Mice were vaccinated by intramuscular injection on days 0 and 7 (Figure 4a). On day 14, tail blood samples were collected, corresponding to 1 week after the final immunization, to assess antibody levels (Figure 4a). Remarkably, the mRNA-LNP vaccine incorporating TMSB10-UTR demonstrated a notable increase in specific immunoglobulin IgG levels against SARS-CoV-2 strains Delta B1.617.2, alpha B.1.1.7, and Omicron B.1.1.529 (Figure 4b). To compare the cellular immune responses activated level between R27-UTR and TMSB10-UTR, we assessed the production of interferon-γ (IFN-γ) and interleukin-4 (IL-4) in the spleen and lymph node using an ELISpot assay on the 28th day post-immunization. Splenic lymphocytes and lymph nodes from mice immunized with the TMSB10 UTR exhibited a significantly higher level of IFN-γ and IL-4 secretion compared to those immunized with the R27 UTR (Figure 4c–f). Additionally, the TMSB10 UTR-mRNA vaccine elicited a greater abundance of CD4^+^ and CD8^+^ T cells in splenocytes compared to the R27-UTR (Figure 4g,h). These results suggest that the UTR of TMSB10 can indeed enhance the immunogenicity of mRNA vaccines in the organism.

## 4. Discussion

Dendritic cells are a specialized class of antigen-presenting cells that play a key role in the immune system [20]. In this paper, we screened the nine genes with the highest mRNA abundance in each dendritic cell subtype (Figure 1). Among these genes, TMSB4X, B2M, and TMSB10, namely, which exhibited a high expression in dendritic cells, were ultimately selected (Figure 2). Notably, the UTR of TMSB10 demonstrated a comparable efficacy in antigen-presenting cells to the previously reported optimized R27-UTR, particularly in DC2.4 cells (Figure 3a), indicating that the UTR of TMSB10 exhibits a greater efficiency for DC cells. Subsequent animal experiments revealed that the TMSB10 UTR demonstrated elevated expression levels in the bloodstream. Given that TMSB10 was initially identified as a highly expressed gene in peripheral blood cells, its UTR performance corresponds well with its source (Figure 3b). Subsequently, we applied it to the mRNA vaccine against COVID-19. The experimental results further demonstrated that the UTR of TMSB10 indeed elicits a higher humoral immunity (Figure 4b) and induces stronger T-cell immune responses (Figure 4c–g). These findings underscore the feasibility of our approach targeting DCs to design UTRs for enhancing their antigen-presenting capabilities. That is not only applicable to SARS-CoV-2, but also to other mRNA vaccines, thus offering broad prospects for application.

However, these natural UTRs, which have the highest mRNA abundance in antigen-presenting cells, still have room for further improvement. For instance, the TOP motif, a cis-regulatory RNA element, initiates immediately following the m7G cap structure and features a characteristic invariant 5′-cytidine, followed by a continuous stretch of 4–15 pyrimidines [21]. It is known to negatively regulate mRNA translation [22]. Upstream open reading frames (uORFs) within the 5′UTR region also serve to suppress downstream protein expression [23]. Potential enhancements could include the deletion of the 5′ Terminal OligoPyrimidine (5′ TOP) motif or upstream open reading frames (uORFs) in the 5′UTR. Another strategy involves incorporating aptamers into the 5′UTR sequence to recruit translation-enhancing proteins or cap-binding proteins, thereby boosting the expression levels of downstream genes [24]. Additionally, strategies could include deleting binding sites in the 5′ and 3′UTRs for microRNAs that degrade mRNAs or inhibit their translation [25,26], or reducing unstructured sequences within the 3′UTR sequence, among other approaches, all of which have the potential to further enhance the protein translation efficiency [27]. Alternatively, we can employ AI for genetic evolutionary training using our existing library of highly expressed UTRs to further enhance their expression capabilities [28]. In addition, the UTR can have a profound effect on mRNA stability, translation, and localization by interacting with various cellular components such as RNA-binding proteins, microRNAs, and RNA structural units. To improve the specificity of UTR targeting to DC cells, we can also add DC-cell-specific regulatory elements to the UTR according to the characteristics of RNA-binding proteins in DC cells. These improvements provide new strategic directions for improving mRNA vaccine efficiency and persistence.

Furthermore, TMSB10 is identified as a protein potentially implicated in regulating cell migration through its actin-monomer-binding activity. It has been linked to the activation of SARS-CoV-2 and VEGFA-VEGFR2 signaling pathways in both extrafollicular and follicular B cells (https://www.genecards.org/cgi-bin/carddisp.pl?gene=TMSB10&keywords=TMSB10, accessed on 12 January 2024). According to our experimental results, TMSB10 might play a significant role in the body’s immune response; yet, it has been understudied, warranting further investigation.

Our study also has limitations. We did not compare our UTRs with the already marketed Moderna and BioNTech mRNA vaccines’ UTRs. This absence of comparative analysis limits our ability to gauge the relative efficacy and potential advantages of our identified UTRs in relation to those already established in commercial vaccines. However, our aim was to propose a strategy for screening natural UTRs for application in mRNA vaccines to enhance antigen immunogenicity. Therefore, we chose the already reported optimized natural UTR, R27, as a control to validate our strategy. In our next work, we will continue to optimize the UTRs we screened and compare them with those in mRNA vaccines already on the market. Furthermore, in this study, we only explored the mRNA expression abundance of the TMSB10 gene in antigen-presenting cells in peripheral blood. Whether it has the same high mRNA abundance in other cell types or other tissues, and whether its UTR can be applied to other gene therapy fields, still needs further research.

## 5. Conclusions

In conclusion, our study identified a pair of natural untranslated regions (UTRs) that significantly augment the antigen gene expression in dendritic cells. This discovery holds considerable potential for augmenting the efficacy of mRNA vaccines, notably against SARS-CoV-2, as well as a spectrum of other pathogens. Nevertheless, it is crucial to recognize the limitations inherent to our investigation. While the results indicate promising avenues for the application of these UTRs in vaccine development, comprehensive studies are imperative to thoroughly understand their mechanisms of action and assess their safety and efficacy over prolonged periods. In addition, the screened UTRs also need to be compared with more reported dominant UTRs for further optimization. Despite these constraints, the inherent potential and adaptability of these UTRs in the realm of vaccine innovation cannot be overstated. Their significance is underscored by the pressing need for more effective vaccines, which positions them as a subject of paramount importance for further research and development.

## 6. Patents

This work has been filed for a Chinese patent under Patent Application No. 202310557980.4.

## Figures and Tables

**Figure 1 vaccines-12-00432-f001:**
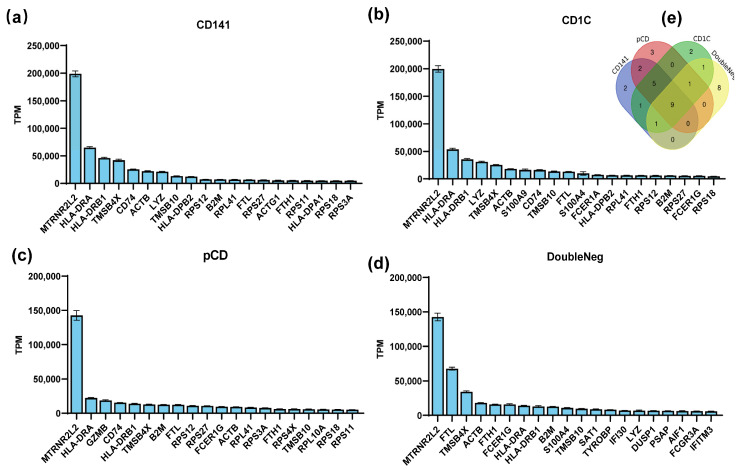
Screening for preferentially expressed genes in dendritic cells (DCs). (**a**–**d**) The GSE94820 single-cell sequencing dataset of peripheral blood mononuclear cells was selected from the GEO database, and the gene expression abundance in each cell subset was analyzed and sorted. The top 20 most abundant genes in each DC cell subtype are shown in the figure. (**e**) Venn diagrams showing numbers of genes that were all expressed within the top 20.

**Figure 2 vaccines-12-00432-f002:**
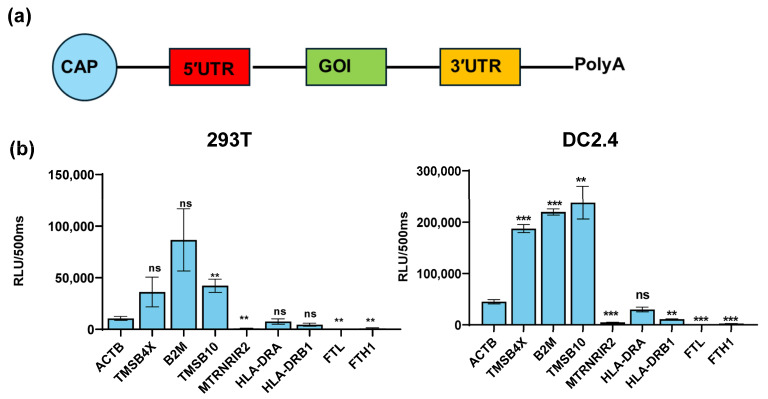
Expression efficacy of different UTRs in DC2.4 and 293T cells. (**a**) Schematic representation of construct for different UTRs. Untranslated regions (UTRs) were incorporated upstream and downstream of the gene of interest (GOI). All mRNAs were synthesized using pseudo uridine and capped by Cap 1 Capping System (Circle). (**b**) Luminescence of different genes’ UTR was measured at 24 h after Gaussia luciferase transfection of 293T cells and DC2.4 (n = 3). All data are shown as mean ± SEM and analyzed using unpaired *t*-test, ns, not significant, ** *p* < 0.01, *** *p* < 0.001, compared with ACTB’s UTR.

**Figure 3 vaccines-12-00432-f003:**
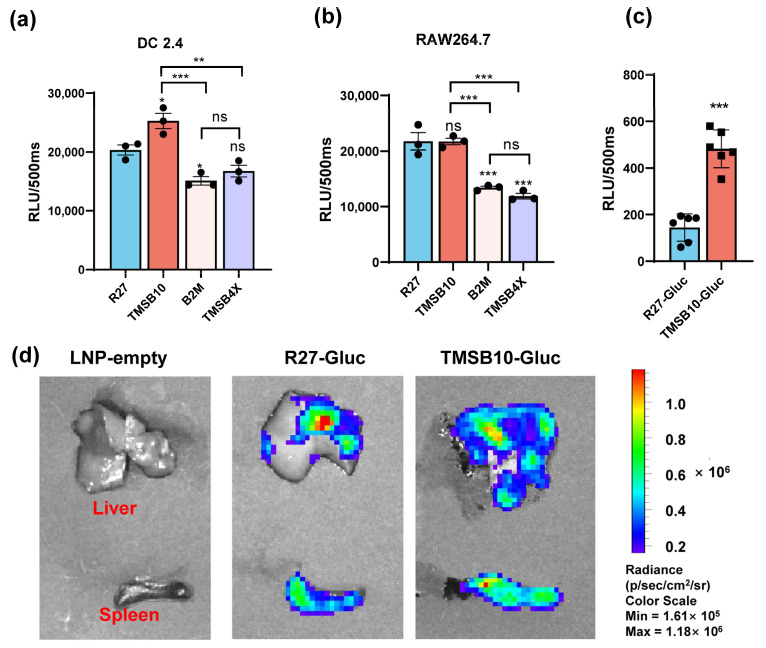
Expression efficacy of dominant UTRs in vitro and in vivo. (**a**,**b**) Luminescence of dominant UTR groups was measured at 24 h after they were transfected in DC2.4 cells (n = 3 (**a**)) and RAW264.7 cells. Data are shown as mean ± SEM and analyzed using one-way ANOVA with multiple comparisons tests. ns, not significant, * *p* < 0.05, ** *p* < 0.01, *** *p* < 0.001. Blue represents group R27, red represents group TMSB10, yellow represents group B2M, and purple represents group TMSB4X. (**c**) Expression efficacy of TMSB10-UTR and R27-UTR in mice blood. Female BALB/c mice were inoculated with 5 ug luciferase mRNA-LNP by intramuscular route. Six hours later, tail blood was taken to detect its luminescence (n = 6). Data are shown as mean ± SEM and analyzed using unpaired *t*-test. ns, not significant, *** *p* < 0.001, as compared with R27-GLuc. Blue represents group R27-Gluc, red represents group TMSB10-Gluc. (**d**) Expression efficacy of TMSB10-UTR and R27-UTR in mice liver and spleen.

**Figure 4 vaccines-12-00432-f004:**
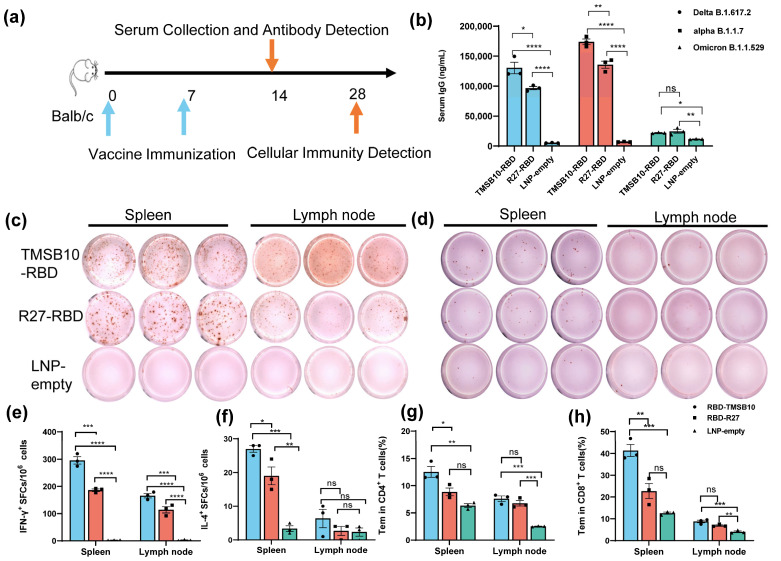
TMSB10-UTR induces higher SARS-CoV-2 RBD-specific humoral and cellular immune responses against SARS-CoV-2 antigen. (**a**) Schematic diagram of immunization, sample collection schedule. Blue arrows represent the time of vaccine immunization, orange arrows represent the time of serum collection as well as the time of cellular immunity testing. (**b**) Concentration of SARS-CoV-2 (Delta B1.617.2, alpha B.1.1.7, Omicron B.1.1.529); specific IgG antibody was determined by ELISA (n = 3). Blue color represents DeltaB1.617.2 strain, red color represents alpha B.1.1.7 strain and green color represents Omicron B.1.1.529 strain. (**c**–**f**) ELISPOT assay for IFN-γ (**c**,**e**) and IL-4 (**d**,**f**) in splenocytes and lymph node cells (n = 3). (**g**,**h**) SARS-CoV-2 RBD-specific CD4^+^ (**g**) and CD8^+^ (**h**) Tem cells (CD44+ CD62L-) in splenocytes and Lymph node cells were detected by flow cytometry (n = 3). All above data are shown as mean ± SEM and analyzed using one-way ANOVA with multiple comparisons tests. ns, not significant, * *p* < 0.05, ** *p* < 0.01, *** *p* < 0.001, **** *p* < 0.0001. Blue represents group RBD-TMSB10, red represents group RBD-R27, green represents group LNP-empty.

## Data Availability

The data that support the findings of this study are available from the first author upon reasonable request.

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
