# Peer review of "Enhancement of SARS-CoV-2 mRNA Vaccine Efficacy through the Application of TMSB10 UTR for Superior Antigen Presentation and Immune Activation"

_vaccines, 2024, doi:10.3390/vaccines12040432_

Round 1

Reviewer 1 Report

Comments and Suggestions for Authors

Since the pandemic, mRNA vaccine platforms have gained a lot of attention. Researchers have been studying different ways to increase the stability and breadth of the mRNA vaccine-induced immune responses. Ding et al. have studied “Enhancement of SARS-CoV-2 mRNA Vaccine Efficacy through the Application of TMSB10 UTR for Superior Antigen Presentation and Immune Activation.”

Although the concept of utilizing optimal UTRs in mRNA vaccine design is not novel, the authors have tried to use the dendritic cell mRNA UTRs in a unique way.

This reviewer has the following suggestions and questions:.

I have a naïve question regarding the proof-of-concept. In general, for vaccine studies, we need a very small amount of protein. It is said that the amount of protein available for priming may have little effect on the recall memory, which will occur only after the booster dose. If this is the case, how will the increase in the expression system influence immunogenicity, particularly the adaptive immune response? I mean, the priming response could be enhanced by changing the concentration; once class switching and T cell polarization happened, there would be very little amount of antigen required for the recall memory. Therefore, this strategy would answer this question.

Lines 58–60: Upon I.M. injection, neutrophils and monocytes are the first migratory cells to the damaged tissue for foreign material. Monocytes become mature once they receive signals from neutrophils. Later, DCs role has to be started, including antigen capture, processing, and presentation. LNP internalization into DCs should be elaborated.

Section 2.1: Details of the construction of the plasmid are missing. How the polyA was added to the mRNA sequence.

Section 2.3: Please provide the Gaussia Luciferase sequence in the supplementary information. Have authors done the codon optimization of this sequence?

Section 2.4:

  • Could you please clearly explain the “N/P ratio of 6:1”? This is very important, and several readers are interested in this aspect.
  • T-tube method: please provide the details of the method, including instrument details as well.

Section 2.6:

  • Why was the 7-day protocol followed? In general, 14 days is the best protocol for mRNA vaccines.
  • 15 µg of mRNA is too much, as 0.1 µg of mRNA would also give wonderful results. In addition, 30 µg is the human dose used during the pandemic.
  • 200 µl I.M. injection. Is this normal for mouse studies? I know only 50 µl is the highest we can use for mice, 100 in the case of two thighs.

Line 157: What antibody is conjugated to "APC/CY7"?

Section 2.9: “Students t-test” The students t-test was used for the multi-group comparison. I am a bit worried about the stat part.

Figure 3c: Why the authors have not collected the image at once. The image looks like the samples were collected individually.

Line 230: “TPA” tissue plasminogen activator??

Figure 4b: Are you sure about the y-axis labelling? 100k ng/ml???

Figure 4g and h: Why the scattered graphs show only one animal.

Author Response

Dear Reviewer,

We extend our sincerely appreciate to you for the constructive feedback, which has been instrumental in enhancing the caliber of our manuscript. The comments have been meticulously addressed as delineated below, with the queries presented in italicized text for clarity. Our corresponding responses are articulated in standard font, whereas modifications and additions implemented in the manuscript are distinctly marked in red.

Sincerely,

Jintao Li

College of Basic Medicine

Third Military Medical University

Gaotanyan street, No.30

Shapingba district

Chongqing 400038, China

Tel: 1-86-23-68771389

Fax: 1-86-23-68771391

E-mail address: [email protected]

Reviewer 2 Report

Comments and Suggestions for Authors

Vaccines- Review 17Mar 2024

TITLE

 Enhancement of SARS-CoV-2 mRNA Vaccine Efficacy through 2 the Application of TMSB10 UTR for Superior Antigen Presentation and Immune Activation

Background

The project was conducted by a highly qualified research group in China and the manuscript although highly technical is well written and I believe will be of interest to the readership. Stabilization and enhancement of vaccine efficacy coupled with safety is a continued urgent need within vaccinology. The authors have taken a highly innovative approach by focusing on the dendtric cell- the master regulator of upstream antigen processing and presentation to other immune mediating cells. They provide a well described and referenced description to support and develop their model to stabilize mRNA and enhance translation within the DC as their target utilizing the UTR of the TMSB10 gene for the RBD of the SarsCov-2 virus. This approach demonstrated relative enhanced immunogenicity towards viral infection.

Material/ methods and design

All described in sufficient detail and utilizing up to date techniques and protocols. Sources for supplies are listed and are acceptable molecular – cell -immunology sources.

Animal studies

Noted as approved by their local IACUC committee and follow standard immunology procedures.

To authors-

1. How many mice per group and was the grouping powered and at what level?

2. Were other antibodies determined specially IgA in serum or other sites such as mucosa?

Results

To authors-

1.     What is the significance of transfection of 293T compared to the DC model DC2.4?

2.     The following observation from fig 3

“Our results demonstrated that TMSB10-UTR-Gluc exhibited higher expression levels in mouse macrophage line RAW264.7 cells, while in 16HBE cells, 204 TMSB10-UTR-Gluc expression was lower compared to R27-UTR-Gluc (Figure 3a).

The reviewer does not see that the TMSB10- had “higher expression” in the RAW cell compared to the R27 UTR in RAW? – please clarify.

Additionally, the authors state-

“These findings indicate that TMSB10-UTR, while comparable to R27-UTR in expression, possesses a cell-specific advantage over R27-UTR in antigen-delivery cells.”

The reviewer sees a similar expression level with R27-UTR versus TMSB10-UTR in the RAW cells (fig 3a). Please clarify.

“ Upon dissecting the mice, we observed that TMSB10-UTR exhibited higher expression levels in the spleen and liver compared to R27-UTR- Gluc (Figure 3c).”- As there is concern for effects in other organs were other organs evaluated at necropsy such as heart, liver and kidney? Were the lungs examined?

Fig 4 results are very compelling and support the proposed hypothesis and aims of the authors. One awaits actual viral infection studies in appropriate animal models that mimic the human manifestations of COVID-19 infection

Comments on the Quality of English Language

Minor editing of English language required

Reviewer 3 Report

Comments and Suggestions for Authors

The article tries to represent a new approach to a SARS-CoV-2 vaccine; however, the model is not convincing starting from the use of RAW cells, and the production of responses in mice may be potentially useful, but it is not novel it does not induce s significant and specific response and hence no potential benefits are observed. Also, the authors did not compare to other well-known systems. More experiments and results are required. 

There are no clear-cut comparisons with other constructs. The results in the first figure only provide a mild response to the construct, which was supposed to be specific, but no valid experiments were performed. The assumption that the construct is inducing a protective immune response was not validated; it was assumed by indirect evidence, and this issue is invalid. The discussion is based on non-proven elements, therefore I had to reject the manuscript. Thus, it is not novel, methodologically sound, and does not contain any relevant results, and the discussion is out of focus.  

Comments on the Quality of English Language

Several sentences require revision.

Round 2

Reviewer 1 Report

Comments and Suggestions for Authors

The authors have revised the manuscript by considering the reviewers' comments. 

Author Response

Thanks very much for your kind work and positive suggestions of our paper.

Reviewer 2 Report

Comments and Suggestions for Authors

I have reviewed the thoughtful comments to my review and find no additional concerns. I congratulate the authors on an exciting study and hope they pursue additional studies to advance this approach in Immunology. 

Comments on the Quality of English Language

Overall excellent 

Author Response

We are very grateful for your kind appraisal.

Reviewer 3 Report

Comments and Suggestions for Authors

The authors did not respond to the queries raised; they just made minor changes in the text. Therefore I do not consider this manuscript suitable for bublication

Comments on the Quality of English Language

Minor mistakes were encountered

Round 3

Reviewer 3 Report

Comments and Suggestions for Authors

The authors made some changes that partially responded to the queries. More experiments were needed.  However, as suggested, they did not include the limitations of the experiments. 
